# Private Animal Welfare Standards—Opportunities and Risks

**DOI:** 10.3390/ani8010004

**Published:** 2018-01-02

**Authors:** Frida Lundmark, Charlotte Berg, Helena Röcklinsberg

**Affiliations:** 1Department of Animal Environment and Health, Swedish University of Agricultural Sciences, P.O. Box 234, SE-53223 Skara, Sweden; lotta.berg@slu.se; 2Department of Animal Environment and Health, Swedish University of Agricultural Sciences, P.O. Box 7068, SE-75007 Uppsala, Sweden; helena.rocklinsberg@slu.se

**Keywords:** animal welfare, audit, compliance, inspection, legislation, official control, private market, private standard, regulation

## Abstract

**Simple Summary:**

In all European countries, farmers keeping animals must comply with European and national animal welfare legislation. Each government has a responsibility to make sure that the legislation is complied with by the farmers. However, during the last decades it has become increasingly common that private organisations, such as the industry, farmers’ organisations, or animal welfare organisations, develop additional animal welfare regulations (‘private standards’) that the farmers also need to comply with. These private standards have the opportunity to improve animal welfare above the legislative level, however, in our study we have shown that this is not always the case and that all of these different private standards, in addition to the legislation, makes it difficult to get an overview of the animal welfare regulation and control arena. For the sake of the farmers, the animal welfare inspectors, the consumers, and the animals we conclude that it is important that policymakers consider this arena as a whole and not their own regulation as a single phenomenon.

**Abstract:**

The current shift moves the governance of animal welfare away from the government towards the private market and the consumers. We have studied the intentions, content, and on-farm inspection results from different sets of animal welfare legislation and private standards with an aim to highlight the most important opportunities and risks identified in relation to the trend of increasingly relying on private standards for safeguarding or improving farm animal welfare. Our results show that different focuses, intentions, animal welfare requirements, inspection methods (i.e., methods for measuring and evaluating the compliance with a regulation), and inspection results, together with the use of vague wordings and a drive towards more flexible regulations does certainly not facilitate the interpretation and implementation of animal welfare regulations, especially not in relation to each other. Since farmers today often have to comply with several animal welfare regulations, including private standards, it is important to stress that a given regulation should never be seen as a single, stand-alone phenomenon, and the policymakers must hence consider the bigger picture, and apply the standards in relation to other existing regulations. This is especially relevant in relation to the legislation, a level that a private standard can never ignore.

## 1. Introduction

Many European countries have long histories of animal welfare legislation, with a focus on preventing cruelty and poor animal welfare by stipulating a minimum level of housing and management. Hence, one of the main aims of any animal welfare legislation is to reduce welfare risks and work with the prevention of poor welfare [1], meaning that if the animals are exposed to unnecessary suffering the animal welfare efforts have failed [2]. In addition, various types of voluntary animal welfare standards have lately been introduced in Europe as a mean to improve animal welfare. One can even talk about a trend of the 21st century of moving the governance of food safety and animal welfare away from government to the market place and consumers [3,4,5,6]. Hence, today, European animal farmers do not only have to meet animal welfare legislation, but also certain kinds of private animal welfare standards in order to be able to market their products [4,7,8]. Private standards may comprise input requirements similar to those that are commonly seen in the legislation, but often also put considerable emphasis on animal-based output variables. These private animal welfare standards have been initiated by different stakeholders in the food chain, including the processing industry (slaughterhouses and dairy plants), primary producers’ organisations (farmers’ organisations, organic farming organisations), retailer chains, and various other governmental and non-governmental organisations [2,8]. There are several reasons why such standards are developed [9], but one general reason for food businesses to establish standards is to show that the products they produce, handle, and sell are of a high quality and safe [4]. Bock and van Huik [10] reported that the main incentive for farmers to participate in any voluntary standard was to get better payment for their products and improved access to the market. However, they also concluded that farmers affiliated to an organic standard or to a specific animal welfare-focused standard were mainly motivated by ethical concerns and the possibility of improving animal welfare.

The current trend to shift responsibility for animal welfare away from the public sphere into the private arena is in many cases supported both by politicians and researchers [11,12], but also evokes a number of practical and ethical issues [3,13]. For example, the presence of both public and private regulations has changed the pattern of inspections of animal welfare at farm level. According to EU legislation, the official control must be risk based (Reg 882/2004/EU) and for example in Sweden membership in a private standard has been included into the risk classification system by the Board of Agriculture. The assumption is that farmers affiliated to a private standard are more likely to comply with the legislation, and that they can hence be inspected by the official control system at a less frequent interval.

The ongoing changes and trends on the animal welfare arena evokes a need to scrutinise the effects of having both public and private regulations to comply with, both for the sake of the animals, the animal owners, and the people inspecting the compliance of all these regulations. Within an earlier project we could conclude that policymakers need to consider several gaps in the policy process; gaps between intentions and requirements, gaps between the requirements and measures, gaps between requirements/measures and on-farm assessment as well as between different regulations [13]. In order to ensure a transparent and coherent policy arena these gaps need to be bridged, highlighted, or explained. Departing in the conviction that each regulation cannot be seen as an independent phenomenon, but has to be placed in a larger context when considering both gaps, duplications, and overlaps between different regulations covering the same area, in this case, farm animal welfare, we will here in more detail analyse the risks and opportunities of having both official EU and national legislation and various private animal welfare standards governing the same area of attention.

### Definitions

The word ‘legislation’ in this paper refers to the legal system and the legally binding legislation, i.e., the written law and its decrees concerning animal welfare and protection. The word ‘standard’ refers to all other kinds of regulatory systems, such as assurance schemes, animal welfare programmes, policies, certification schemes etcetera, to which affiliation is usually formally voluntary or market driven. The term ‘regulation’ covers both legislation and standards. The term ‘inspector’ is used for the person carrying out animal welfare inspections, regardless if that person is employed by the official authorities, by a private organisation or by a third party audit provider. In this paper, a ‘requirement’ is the text written in a regulation, i.e., what is prescribed by the policymaker. A corresponding ‘measure’ is then a measurement or an observation taken or made during inspections for controlling compliance with the requirement.

## 2. Materials and Methods

The paper is based on discussions emanating from our project Mind the Gaps! From Intentions to Practice in Animal Welfare Legislation and Private Standards [13], but also on results from other research groups and studies, such as those by More and co-workers [3], who identified a need for greater scrutiny of private standards, and Main and co-workers [14], who have presented a framework for developing private standards with an aim to improve animal welfare. They have, however, not scrutinesed the relation between existing regulations and the effect of the control outcomes. Our project contained three sub-studies that focused on the intentions behind different animal welfare regulations [2,15], the content, and structure of different animal welfare regulations [16,17] and similarities and differences between on-farm assessment and inspection results based on different regulations [13].

The first study involved information from 14 regulations from four different European countries (Sweden, Spain, the UK, and Germany). Results from an EconWelfare questionnaire were used for an overview giving basic information. The questionnaire was then followed by thorough systematic text analyses of the farm animal regulations as well as any preparatory work, explanatory notes, web pages and brochures that were published [2]. The second study was a summative description of four Swedish animal welfare regulations, followed by directed content analyses on the different sets of animal welfare regulations, including any corresponding audit or control guidelines [16]. The third study involved determining the actual outcomes of different animal welfare inspections using the same Swedish regulations as in the second study. Inspection reports were collected and analysed, and the most common non-compliances were identified, risk factors influencing the probability and level of non-compliance were quantified, and, the private standards were compared with respect to their effect on the level of compliance with legislation [13].

Based on these studies, we have identified a need to discuss and scrutinize a number of issues in order to show the complexity of handling the plethora of regulation on animal welfare in a transparent way. A first issue concerns the intention behind a private standard and legislation, which may overlap, but must not, and if so, might lead to complications in the actual implementation. A second issue that is worth discussing is the fact that different regulations tend to focus on quite different aspects of animal husbandry. Whereas, some have a clear focus on animal welfare, others may focus more on food safety or environmental aspects. Such differences may not always be obvious to neither farmers nor consumers. One of the main reasons given for revision of regulation and opening up for private standards is a demand for increased flexibility in compliance, and since this implies a risk of lack of transparency and consistency, this is an important third issue to scrutinize. The fourth issue that we elaborate on is the shift from governmental to private regulation in terms of an implicit transformation from regarding animal welfare a public good to a private good. This evokes a need to discuss implications of shifting focus from citizens to consumers as well as questions related to dissemination of information.

Our aim is to compare public animal welfare legislation and private standards in terms of intentions, requirements, measurements, and results. Furthermore, our aim is to highlight the most important opportunities and risks identified in relation to the current trend of increasingly relying on private standards for safeguarding or improving farm animal welfare.

## 3. Discussion

### 3.1. Common Intentions and Requirements—Effects on Implementation?

There are many examples of private standards having led to considerable improvements in terms of animal welfare, both in relation to the stipulated requirements and also with respect to compliance [18,19,20]. According to Main and co-workers [14], a private standard’s aim should be to improve animal welfare and not only certify compliance with the governmental minimum legislation. According to More and co-workers [3], two of the first questions for a policy maker to ask and determine is what the private standard’s goal and intention should be when it comes to animal welfare, and who should benefit from the standard. These questions would perhaps be obvious to most of us. However, while scrutinizing private standards it became clear that reality is not that simple [13]. Furthermore, such standards may be quite difficult to compare with other regulations, for several reasons. 

Firstly, we can conclude that private companies generally have the opportunity to improve animal welfare by including requirements in their standards above the legislative level [18,19,21]. Covering a geographic area from US and Canada to Europe, South Africa, and Australia Fulponi [19] reported that 33% of the private standards had some requirements that were significantly higher, and 50% slightly higher than the national legislation. However, such generalisations have to be read with care, as the national animal welfare level differs widely, and affiliation to private standards may also have an impact on farmers’ compliance with the legislation in itself. In countries where animal welfare legislation has been described as limited or the official control insufficient, private standards have been reported to have improved the general animal welfare level, for example, in Brazil [22] and Canada [23]. In our project, there were both private standards aiming at roughly the same level as the legislation and private standards that had requirements on a higher level [16]. Also, Annen and co-workers (2011) found that private standards in Germany often tended to overlap with national legislation, and not necessarily provided better animal welfare, except for the organic standards. In Sweden, the private standard Arlagården is considered a basic quality assurance scheme (according to Bock and van Leeuwen’s [24] definition), primarily equal to national legislation. However, if exactly the same version of the Arlagården standard was implemented in another country it would most likely be considered as a top quality assurance scheme, if that country had a lower legislative standard for animal welfare than Sweden. Hence, a top scheme in one country may be considered a basic scheme in another; it all comes down to what legislation that you compare it to. According to Main and co-workers [14], this lack of standardisation does complicate international trade, as there is a lack of clarity with regards to the animal welfare levels. Furthermore, More and co-workers [3] raised similar concerns about no consensus between the multiplicity of private standards. However, they also concluded that standardisation may not be in the interest of the policy makers, since they want to illuminate differences with respect to products that are otherwise similar.

Mullan and co-workers [25] reported an animal welfare improvement in non-caged laying hens in the UK when farmers were affiliated to the RCPCA or Soil Association standards, i.e., a generally positive animal welfare effect of such affiliation. There are indications both from our project [13], but also from others [20,26], that an affiliation to a private standard may lead to better compliance with the legislation. However, the welfare levels may also be lower in some respects on farms affiliated to private standards. For example, Main and co-workers [27] reported that Freedom Food-affiliated farms did have more lame cows and also more cows experiencing severe raising restrictions when compared with farms that were not affiliated to Freedom Food, but fewer problems with mastitis and dirty cows. Thus, even if private standards can have the opportunity to improve animal welfare and increase the compliance with the legislation, this is a complex matter that needs to be investigated further.

Secondly, we have found that several regulations often seem to have similar intentions and use the same central concepts even if the detailed requirements differs significantly [2]. This clearly hampered any comparison of the regulations. Two concepts that were often used as central themes in the regulations were; ‘unnecessary suffering’ and ‘natural behaviour’. The notion of (un)necessary suffering or natural behaviour was not defined in any of the regulations and it was clear that different initiators made different decisions as to whether an action or situation would result in suffering, and had different views on what kind of suffering was considered unnecessary [2,15]. After examining procedures known to cause pain (e.g., beak trimming, castration, and slaughter without prior stunning), and therefore suffering, it has become clear that the different policymakers behind the regulations have different views regarding what is unnecessary or not. All policymakers basically have access to the same scientific knowledge about animal sentience, physiology, and welfare, but have drawn different conclusions with respect to what is considered to be necessary or important, which is then mirrored in what is legal or acceptable. Culture and tradition, economics, consumer demands, food quality, ethics, and religion all play a role when deciding if something is unnecessary or not, as is the case for the general aspects of how animals can be kept and used by humans [2,28,29].

Providing the animals with possibilities to behave naturally is a common theme in animal welfare legislation [30,31,32], but is understood in different ways. Interpretations range from striving to let animals live in a way resembling their wild ancestors’ lives to accepting a confined indoor environment as long as it allowed for animals to perform only some crucial behaviours [2]. Furthermore, the term was usually used as a rather narrow concept in the regulations focusing on the design of the animals’ husbandry system and environment, rather than on social structures, weaning, feed, mutilations, or mating behaviours. It is probably strategically smart to include the ability to perform natural behaviours as a claim when trying to appeal to consumers and to satisfy citizen requirements, since a natural life is important for lay people [33], but consumers are seldom aware there is a limitation as to what natural behaviours are granted. The different use of the same central concepts also indicates differences between the meanings of the concept of animal welfare; and, where the emphasis should be—on the animals’ health, subjective experiences, or on behaviour, or all three.

Thirdly, other risks related to the interpretation and comparison of the regulations are the presence of different vague wordings (‘satisfactory’, ‘dignity’, ‘appropriate’, etc.), and when a private standard claims to aim at the same animal welfare level as the national legislation but has expressed the requirements differently. Does for example a Swedish dairy farmer have to give the same amount of straw to the cows when trying to comply with different regulations; ‘enough straw’, ‘an adequate bedding’, ‘a bed of straw’, and ‘generous amounts of litter’? Does this difference in wording indicate an actual difference in what is actually required? Hence, it was not always clear if the requirements in the regulations were practically identical or whether there was a difference [16]. Similar conclusions regarding ambiguities in animal welfare regulations have been identified also by others [29,34,35,36,37]. Unclear words and vague concepts in a regulation increase the risk that different opinions will evolve about how to interpret and implement these in practice [34,38], leaving the determination of compliance entirely up to the individual inspector or to unwritten rules or informal practice within a team of inspectors. It is important to realise that vague expressions does not only give room for flexibility, but may also lead to uncertainties, both within regulations, and between them.

We argue that above vague wording, it is often the implicit intentions behind the regulations that generate these differences and risks. During the development of both legislation and private standards, the animal welfare level is adjusted to be in line with other areas of interest, e.g., ensuring a strong and competitive agricultural sector, environmental concerns, food safety, financial limits, and practical constraints. Such value clashes are inevitable in farming, but we argue that if a regulator avoids drawing clear cut lines between what is accepted and what is not in terms of resources and management, or factual animal well-being, this may instead create a need for farmers and inspectors not only to interpret vague formulation of regulation, but also to handle such value clashes that lie behind the standard in the actual compliance on farm, or even animal, level. This is not necessarily negative per se, but if the background, education, and preconceptions of an inspector are allowed to largely influence the inspection results, which may then vary considerately between inspectors, this may be perceived as unpredictable by the farmer, and may in severe cases lead to legal injustices. This situation is related to the next issue that we want to mention.

### 3.2. Different Focuses between Regulations 

It is sometimes mentioned that private standards, which include several areas of interest, have the opportunity to be more successful in dealing with animal welfare, reaching out and appealing to a higher number of consumers, since there is a risk that consumers get ‘worn out’ if the focus is on animal welfare only, or that their top priority is not animal welfare, but instead, for example, food quality [11,39,40]. In line with this, we have found that several of the private standards involving animal welfare also covered other areas, such as food safety, food quality, and environmental issues. As a matter of fact, neither the initiators of the animal welfare legislation nor the private standards were actually considering only animal welfare. Instead, all of the regulations were partly based also on several other factors, such as economy, culture, traditions, religion, consumer demands, environment, food quality, food safety, disease risks, etc [2]. These factors were not always obvious in the regulations themselves, but implicitly or explicitly mentioned in documents concerning the policymakers’ intentions and motives. Also, other studies have shown that not just animal welfare science has an impact on decisions related to animal welfare regulations [41,42,43].

The fact that the regulations’ focus on animal welfare differed might explain why the on-farm inspection results differed between regulations [13]. For example, in the county of Västra Götaland, in Sweden, the most common non-compliance in dairy farms according to the official animal welfare control was dirty cattle, while instead, the most common animal welfare non-compliance of Arlagården was dirty stables/cowsheds [13]. This could be explained by the focus that Arlagården has on food safety and hygiene, which is not the main focus during official animal welfare inspections. Such differences in focus, even if the regulations’ actual requirements appear to be similar, increase the risk of different on-farm inspection results, even if the inspections are made simultaneously. These different results may be confusing for the farmer unless the inspectors have been very clear about the focus of their own inspection, and possibly also why the results might differ. Furthermore, these differences also stress the fact that statistics from animal welfare inspections will not at all provide a complete picture of the welfare level in a region or country, although it has been suggested that such statistics can be used for monitoring the nature and magnitude of welfare problems [44]. Inspection results naturally mirror the compliance with a certain regulation, not an ‘objective’ general animal welfare level, and hence the results from the inspections related to different regulations can differ, even if the same animals are inspected, and hence the welfare level is identical.

### 3.3. Using Different Measures to Assess Animal Welfare 

Since it is difficult to establish an ‘objective’ description of the complete welfare state of an animal, the use of different assessment protocols within the legislation and various private standards may in general increase the number of aspects covered, and thereby facilitate creating a more comprehensive picture of the animal welfare status of the farms involved. There is, however, also an inherent risk of confusion for the farmer receiving different on-farm assessment results related to one and the same farm, if he or she is not fully aware of and knowledgeable about how the requirements are to be measured [16], i.e., on the choice of indicators and control points. We found that the control guidelines corresponding to the regulations could differ even when the requirements stated were similar. Such differences could be seen on three levels. (A) A number of different measures could be corresponding to one particular requirement. This means that for the same type of requirement there could be different number of indicator variables to be used. Sometimes the number of measures to be checked at an inspection substantially outnumbered the requirements, leading to the perception that additional requirements were introduced in the inspection guidelines, i.e., requirements that had not been clearly stated in the regulation. (B) The measures could also be of different types. For example, a requirement expressed as a resource-based requirement could in the inspection guidelines be suggested to be measured also in other ways, using management- and animal-based measures. (C) Finally, the inspection results could differ as a result of whether the animal-based measures were taken at the herd level or at the individual animal level. The Swedish legislation, as well as in other countries (e.g., Norway, Finland, and the UK), has an individual-focused approach to the welfare of animals [29,45,46]. If one animal is in poor condition this is non-compliance, and corrective action must be taken. However, some of the private standard recommended measuring of animal welfare outcomes at the herd level; hence, non-compliance will not be reported until a certain proportion of the animals were affected. This approach means that the welfare of one animal, and the correction of welfare problems identified, is dependent on the welfare of all the other animals at a farm. This view not only is a significant step away from the legislator’s emphasis on the relevance of each individual animal, but one should also keep in mind that an animal in poor condition does not suffer less just because the assessment at farm level indicates an overall high welfare level in the herd.

Already in 1980, Dawkins concluded that the measures that mirror productivity, e.g., mortality, are unreliable indicators of animal welfare because they are applied to a group of animals rather than to individuals [47]. Since animal welfare is by definition a characteristic of the individual animal [48,49] and the basic general animal welfare legislation aims at protecting individuals, it seems contradictive to measure welfare only at the herd level when evaluating compliance. We argue that as long as the legislation cares for individual animals, group assessments cannot replace individual assessments. Having said this, measuring animal welfare at group level can still be a useful complement aimed at identifying systematic problems (e.g., why a certain proportion of the animals in a herd is lame, dirty, suffering from mastitis etc.), to make improvements that could contribute to preventing poor welfare for future animals at a given farm.

Hence, even if similar requirements were stated in the text of the legislation and private standards, differences were found both regarding methods of assessment and as to when compliance was considered to be achieved or not. It can actually be questioned if these private standards do not in fact sometimes offer a protection level that lies below the legislative level, as severe problems that are related to a low number of individual animals can be completely ignored. We argue that this finding is quite remarkable and that such a standard, or implementation of it, will not benefit the animals, the farmers, or the animal welfare arena as a whole. Rather, it might evoke the opposite of the intention of formulating and affiliating to it, i.e., lead to lower consumer interest in animal welfare issues, and as well as decreased trust in (implementation of) standards and legislation. If consumers feel cheated upon, they might either stop purchasing animal products all over, or let price be the only criterion, neither of which would benefit animals or farmers affiliated to various standards, or the consumer/citizens may be prompted to ask for further regulation. Our findings also show that when comparing regulations, it is important to not only compare the requirements, but also the corresponding inspection measures and control procedures, to safeguard that necessary information about the regulations is not lost or ignored.

### 3.4. Flexible Regulations 

One opportunity that is often mentioned in relation to private standards is that they can be more flexible than the legislation, as they can easily be changed, updated, and adapted to new circumstances, developments within a sector, and research results, while the development of legislation is often perceived as a slow and bureaucratic process [6,7,18,21,50,51]. Quite expectedly, we can also within our project conclude that the private standards were updated more often. The question is, however, if it is always an advantage to have a regulation that can easily and rapidly be changed? 

Firstly, when making amendments and updates it is important to make these well-known to the farmers and other animal owners who have to comply with the regulation in question, or there is a risk for non-compliances at the next inspection, based on ignorance about the changes made [13]. Since several of the private standards are in practice mandatory for a farmer to comply with in order to have access to the market, it is important that also the private standards, not only the legislation, are striving for legal predictability and transparency, which puts a high demand on the initiators, for information transfer. 

Secondly, another type of flexibility mentioned as an opportunity linked to private standards is that they can be more goal-oriented than the legislation, i.e., consist of animal-based requirements describing what to achieve (in animal appearance) and not what kind of management and resources (input) that is needed, stimulating individual farmer innovation to find different types of solutions to improve animal welfare outcomes [52]. This kind of flexibility in animal welfare legislation has, however, also recently been requested by the European Commission [53]. Anneberg and co-workers [54] found that Danish farmers would like to be treated more individually during inspections, and simultaneously, they also wanted the inspections to be more standardised and treat all farmers equally in order to ensure predictability and transparency. Based on our analyses, we argue together with Wahlberg [29] that from an animal welfare point of view, it is important that flexible requirements are interpreted from an animal perspective, in accordance with the aim of the animal welfare legislation, or else this flexibility may lead to arbitrariness and de facto lead to poorer animal welfare outcomes. In order to reach a high level of intra- and inter-observer agreement, it is important that the animal welfare goal in itself as stated in the regulation is not overly vague and flexible (which is actually the case in several animal welfare requirements). The goal has to be clear both regarding its content, and when it is achieved; it is only the method to reach the given goal that may vary, not the goal itself, or else the risk for poor legal predictability is obvious. 

### 3.5. Preventive Regulations

Another aspect of flexibility to consider—still in relation to the quest for more goal-oriented regulations—is to what extent the regulation should be preventive. Prevention is perhaps best defined negatively: it has failed when animals are found to be in poor body condition, dirty or soiled, not able to perform natural behaviours, or generally miserable. The relevance of prevention is often discussed in relation to whether emphasis in regulation should be on input or output-requirements, respectively. Input requirements and measures (e.g., resources and management routines), if valid and well-chosen, are important in order to identify and mitigate risks to animal welfare, thereby preventing welfare problems [48,55]. As mentioned above, animal welfare legislation is aiming to protect animals from unnecessary suffering and to reduce welfare risks mainly by setting up input requirements [1,16], which are also considered to be more practical to use and easier to assess than animal-based requirements and measures [56,57]. This means that input requirements generally score high on validity, reliability, and feasibility. On the other hand, animal-based measures will reflect how the animal is actually coping with its environment [58], helping to assess the actual state of welfare for an individual animal at a given point in time, and verify if the preventive efforts have been effective, i.e., resulted in a state of welfare that is aimed for in a regulation. We agree with others [14,38,59] that if a regulation is expected to both act preventively and be used for assessing animal welfare, there will be a need for both resource-, management-, and animal-based mechanisms.

However, in our project, we noticed a difference in the emphasis between the legislations and some of the private standards in relation to the preventive purpose. Even if all regulations claimed to have a preventive purpose, the private standards in Sweden generally consisted of a higher proportion of animal-based requirements than the governmental legislation [16]. Main and co-workers [14] argued that it could be suitable for a private standard to have a more outcome-based approach, when possible and given a level above the legislation, since such an approach may stimulate farmers’ innovation and be used for bench-marking farmers’ performance in order to strive for a continuous improvement-based approach. Furthermore, a higher proportion of resource-related non-compliances were found according to the legislation-based official inspections than according to the private standard inspections [13]. Nevertheless, the main non-compliance according to the legislation-based inspections was dirty cattle, which is still an animal-based measure [13]. Hence, there is possibly a risk that there are differences in focus between inspections related to whether a certain inspection was performed based on a governmental or a private standard check list. As a result, different inspection outcomes may result in different perceptions, by farmers and inspectors alike, of the importance of preventive regulations.

### 3.6. Public Good or Private Good

Traditionally, animal welfare has been considered a public good, a non-competitive issue concerning the whole society [60]. A market driven approach tends to commodify animal welfare as a private good, where consumers seek to satisfy their individual value preference, and in contrast to a public good citizens can’t anymore influence the content of the ‘good’ through the political processes [61]. Private standards (if clearly labelled on the products) are often mentioned as having the benefit of helping to increase consumer concern about animal welfare, enable consumers to buy and consume animal-friendly products, and increase the market share for products ensuring higher animal welfare as a result of consumer demands [18,62]. The shift from state governance towards private governance is based on the assumption that a large proportion of today’s consumers care for animal welfare, and that many consumers are willing to pay for premium products to improve the animal welfare level, i.e., ‘vote with their fork’. Some have even argued that if the national baseline legislation is above the EU level, then the national government should consider lowering the legislation level and instead trust the market to maintain or increase the welfare level through private standards [11]. However, others believe that there is an over-reliance on market mechanisms and consumer behaviours, instead arguing for citizens’ putting pressure on politicians to take the responsibility for ensuring a decent welfare level [61,63]. In this context, we believe that there are a number of risks to be aware of and consider if the market forces are to be given more responsibility for animal welfare in the future. 

Firstly, there are deep concerns that consumers are not knowledgeable enough about modern farming systems and animal welfare to make this well thought through decisions [64,65,66,67], which also makes it difficult to know what standards to trust and what they actually cover. This does of course relate to both private standards and public legislation, and may be perceived as a problem in a wider democratic context. The gap between consumers’ and producers’ knowledge about animal production is sometimes obvious since the average consumers have never visited a production farm, and hence is more and more disconnected from the livestock production [66,68]. Sometimes consumers virtually deny the fact that the meat they eat comes from living and sentient animals that had to be killed before ending up in the supermarket [69]. According to the Eurobarometer, almost two thirds (64%) of the European citizens are however interested in getting more information about the conditions under which farm animals are raised [70], indicating that they believe not having received enough information. In our project we found that private standards (especially when retailer driven) were often focusing on the consumers’ expectations and perceptions, rather than on biological principles and the animals’ needs [2]. A legitimate question for policymakers to ask themselves then, when writing regulations, is whether what consumers consider to be most important to animals is in fact what is actually most important from the animal’s perspective. Hence, there is a risk that a private standard created with the aim of selling a premium product may focus on images and messages that are easy to communicate and appealing to consumers rather than on actual animal welfare improvements.

Secondly, the consumers do not necessarily receive sufficient and correct information about the intentions, content and achievements of private standards. This can also apply to public legislation. Some claim that industry commercials are often misleading, portraying production practices as being much more welfare-oriented or ‘natural’ than they actually are [64,65,71]. Borkfelt and co-workers [64] took the Danish dairy company Arla Foods, the Swedish Poultry Meat Association, and the Danish pig farming as examples when demonstrating misleading commercials from industries managing their own private standards by showing photos of relaxed cows and pigs in a green summer field, or a teddy chicken. This is not chosen arbitrary—studies have shown that consumers value outdoor access and low stocking densities [72,73,74]. Hence, producers’ organisations may have an interest in not showing the normal standard housing and handling routines as that probably would contradict both consumer beliefs about farming and what they want to purchase [64]. Private standards designed to hide or whitewash the realities of modern farming can hence be lacking validity from an animal welfare perspective.

Consumer knowledge is also dependent on transparent regulations. Given the trend that improved animal welfare shall be market driven, consumers need to have access to relevant information about standards, control measures and possibly also the results. It is not easy for the consumers to make well-informed choices if a regulation is confidential [75]. During our study, we were, for example, denied access to the Marks & Spencer standard, to KRAV’s, and Seal of Quality’s more detailed control results from the third party audit bodies that they use [2,13]. This also means that it is impossible for ordinary consumers to actually know the requirements of Marks & Spencer’s animal welfare standard or how the KRAV and Seal of Quality standards are complied with. As reviewed by More and co-workers [3], a key concern related to private standard are the credibility related to private standards being outside the public (governmental) sphere. According to Aerts [75] some retailers are using the standard as a competitive advantage, and are therefore keen on keeping the requirements confidential, and according to Mench [51] a confidential approach with respect to audit outcomes is not unusual since the purpose of private inspections is to provide retailers with information about compliance among their suppliers, rather than to provide consumers with such detailed information. Furthermore, it is not in the interest of the retailer chain to admit that their animal welfare standards are not necessarily always complied with. Since official control includes the exercise of public power against individuals, actions must be taken by the governmental authorities to ensure legal security for individual citizens and provide them with service, support, and information [76]. Private standards are, however, not a part of the legal system and do not have the same requirement of transparency and predictability. We agree with both Main and co-workers [14] and More and co-workers [3], that a trustworthy compliance control is necessary in order for a private standard, as well as for the legislation, to be credible. In our last study [13] we discovered different approaches taken by Arla and by the official control when it comes to the follow-up of non-compliances. While the official animal welfare inspectors mainly made follow-ups on farms, the Arla inspectors mainly had a routine of letting the farmer sending them a written statement that they have reach compliance. How trustworthy different control and follow-up methods are remains to be investigated. However, there is an obvious risk that private standards are less transparent and credible when compared to legislation [50], since private standards do not need to be transparent or decided upon in a democratic process [21]. Nevertheless, private standard operators may choose to disclose the results anonymised or aggregated, should they wish to increase transparency. 

Thirdly, it may be questioned if a high enough proportion of the consumers are willing to pay for premium products [77], even if there are several studies concluding a ‘willingness to pay’ (WTP) a higher price for animal welfare [78,79,80,81]. Even if these studies show a theoretical WTP and a concern for animal welfare, others have shown that consumers do not always act as in accordance with their expressed preferences [67,77,82], i.e., they are not actually buying premium products when shopping. There are several possible reasons behind this inconsistency. One reason may be low average incomes [77]. Another reason may be consumers receiving contradictory information about farming [68], getting an overly positive view of animal production [83] e.g., by advertisement mentioned above, which of course complicates any investigations about consumers’ willingness to pay. According to the most recent Eurobarometer, 59% of European citizens are willing to pay a premium price for animal welfare products [70]. However, the Eurobarometer also found that approximately one third is not prepared to pay more for animal welfare (35%). It should be mentioned that stricter national legislation could in itself lead to higher prices (‘premium products’) in comparison to imported products. It is, however, usually difficult to single out the effect of animal welfare legislation in relation to other national and regional factors, such as labour costs, farm building construction costs in relation to climate differences, feed costs, and the price of agricultural land.

Fourthly, connecting to the point of public and private goods above, consumers do not want to have the sole responsibility for improving animal welfare [63,70,84,85]. In the Eurobarometer study, the results showed that European citizens think that animal welfare is a concern not only for the consumers but for all citizens, including those that are not eating meat, milk, or egg products themselves, and that government must be involved in some way [70]. We argue that the present political drive to stimulate the market to take the responsibility of animal welfare is leading to too rapid and possibly too radical changes, as the future consequences of this transfer have not yet been properly investigated, and that the politicians are hence withdrawing prematurely from their responsibilities. Animal welfare is an area of public concern, where many difficult decisions and trade-offs are needed, of course it is convenient to escape that responsibility and the blame the market and consumers when something goes wrong. 

It is also important to remember that consumer trust in different stakeholders (e.g., policymakers, farmers, and the industry) will differ between countries [67,86], as will also the general legal and public systems. Therefore, it is difficult to find one single solution in the relationship between the public and private sphere that all European consumers will agree upon.

Finally, citizens who are not consuming animal products are not able to affect how animals are kept and treated when the market is solely responsible for the animal welfare level [62]. While the governmental legislation is mainly written based on public concern, and sees animals as a public good, many of the private standards mainly exists as a result of consumer concern [2]. This means that shifting the responsibility from the public sphere to the private area excludes citizens not being consumers, citizens that may actually be non-consumers because of animal welfare reasons.

### 3.7. The Regulations’ Impact on Farmers 

It is important to not forget the farmers’ role on the animal welfare arena. They are the ones that have the daily responsibility for the animals with immense influence on animal welfare, and are the ones that must keep track of all the different regulations and understand the requirements in order to comply with them. Affiliations to private standards is mentioned as an opportunity for the farmers as they gain quality assurance of their production, and the possibility to sell their products at a premium, i.e., at a higher price [87,88]. There are actually very few studies on the experience of farmers with respect to animal welfare regulations, when considering both official and private regulations. However, there are some risks to be aware of.

Firstly, there are indications that some farmers perceive private standards as a necessary evil, i.e., an economic necessity rather than a choice [89], in order to get access to the market [90]. 

Secondly, farmers are generally faced with an increased burden of inspections and requirements, not only related to animal welfare. In our study, we found that although the average interval of official animal welfare inspections is approximately 10 years, many of the dairy farmers were inspected by both official animal welfare inspectors and Arla auditors within the same six month period, and sometimes even had ongoing cases (e.g., series of inspections) overlapping in time [13]. In relation to other studied areas of regulations and advice (e.g., biosecurity and advisory services), Swedish farmers have mentioned problems to keep track of all the different controls that are performed on farm [91,92]. There are also studies from Denmark and Finland showing that farmers do not always understand the meaning and benefit of official animal welfare control [54,93]. However, no such scientific study is, to our knowledge, carried out on the expectations and experience of the whole animal welfare control arena, including private standards and controls. 

Thirdly, farmers are faced with an increased burden of administrative costs, i.e., costs that are related to time spent on paperwork, such as record keeping, and on accompanying visiting inspectors and auditors. More and co-workers [3] raised concern about the compliance costs, and Swedish farmers believe that animal welfare regulations cause a too big administrative burden on the individual farmer [91,92]. Others have mentioned the administrative costs associated with official legislation as a disadvantage for farmers [39]. However, according to our study, the private standards—including the producer initiated schemes—often had more administrative requirements and a higher frequency of inspections, than the animal welfare legislation [13]. This may indicate that farmers do have difficulties distinguishing between the different regulations, and that they may perceive the controls carried out on their farms as a whole and not as single occasions carried out by different control bodies with different focuses and requirements. The farmers may also perceive more of a financial dependency in relation to the private standards, related to access to the market. 

Fourthly, despite the increased burden for farmers they are sometimes offered very little in reward for their efforts [89]. An affiliation to a private standard often entails additional costs (e.g., membership, adjustments to higher requirement, and third party auditing), even if the farmers do not receive a premium [88,89,94]. Of the Swedish private standards in our study, only membership in the organic standard KRAV is linked to a higher price to producers. Dairy products that are labelled with Arla or Swedish Seal of Quality are considered basic levels foodstuffs, i.e., not premium products.

Finally, some argue that the increased power within the private food sector through the implementation of private standards increases the risk of small-scale producers being forced out of key markets, and potentially out of business, as many small-scale producers can have difficulties to meet the volumes required to enter into commercial supply agreements governed by, for example, public procurement processes, or to meet the costs of extra technical requirements and the extra burden of controls [19,90]. More and co-workers [3] raised concerns for private standards to be a discriminatory barrier to trade. 

## 4. Conclusions

The shift in governance of animal welfare from the public arena to the private sphere and market forces does involve a number of opportunities. By having higher goals and demands than the legislation, well-designed private standards may increase consumer knowledge about animal welfare, increase the demand of animal-friendly products, leading to more animals experiencing a higher quality of life. In order not to put such achievements at risk, it is nevertheless important to consider and prevent possible negative consequences of private standards entering the market. Farmers have to be able to handle and understand all these different regulations in relation to each other, regulations that have different intentions, focus, wordings, measuring methods, and animal welfare levels. This also leads to difficulties when it comes to predicting the outcome of an inspection. Furthermore, inspectors—both from private and governmental organisations—have similar difficulties in interpreting and deciding what is acceptable practice. This is even more so if a regulation also requires flexible implementation, since decisions then are even more dependent on individual discernment, and hence a risk for inspector bias, indirect or direct pressure put on the inspector, and lack of transparency in evaluation and decisions. A plethora of animal welfare related standards and labelling schemes would not necessarily be easy to handle for the consumer. The consumers are expected to find knowledge and remember the differences and pros and cons with a number of different regulations, but since some private standards lack transparency, this is at times impossible. Finally, the citizen who does, for any reason, not buy or consume meat, milk or eggs or other animal-derived products risk being left without substantial influence on animal welfare if the governmental institutions withdraw too much from this policy area, which may not be a desirable development.

In summary, a private standard can never be seen as a single, stand-alone phenomenon. The policymakers must consider the bigger picture, ensure transparency, and apply the standards in relation to other existing regulations. This is especially relevant in relation to the legislation, a level that a private standard can never ignore.

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
