# Peer review of "Private Animal Welfare Standards—Opportunities and Risks"

_animals, 2018, doi:10.3390/ani8010004_

Round 1

Reviewer 1 Report

General

- Be clear what is citing earlier papers by the authors

- I am not sure this paper needs a Materials/Methods section and I dont think it added much (and, in any case, there is no results section). The paper needs to be located in the literature, in particular the work cited in this section (especially the authors') but I dont think needs own section

- I think the authors' intellects and previous work places them well to give some stronger arguments on some of the issues (where they do, it is really interesting and useful). Often you raise questions/raise issues to consider, but I think you should go further and answer those questions/suggest how to resolve the issues. Indeed, you seem to imply you have a view (sometimes not that subtly), so be braver. I'd personally suggest making it more a series of mini-arguments (and I've suggested some that could be their own sections), each ending on a prescriptive conclusion. It could then be a great paper on how private standards are best used (and/or how not) ie how to fulfil the opportunities and minimise/mitigate the risks (the paper gets to some of these in the conclusion, but do not give the value of these conclusions justice)

- The paper feels like (though perhaps not - I havent codified it!) it covers risks more than opportunities and could do with a bit of balancing - I think the 'how to do it well' idea suggested above would achieve this

- Since the paper is generally comparing private to public (esp in that section), need to be clear where the risks also apply to public legislation

44 - maybe specify 'unnecessarily' or 'unavoidably' suffering 

68 - should 'have' be 'has' (or membership become plural)?

90 - I dont think regulation is a helpful term here, as might make most readers think only of legal/regulatory settings - whereas the authors mean something wider. Perhaps better  just use 'requirement' (as you do) for both 

121+ - I got a bit lost reading this so suggest a bit of a rewrite (depending on above point re line Methods)

144 - had some requirements (I think)

149 - Do you have a ref that in these countries legislation/control is suboptimal

180-5 - Some greater depth to this point and/or examples would make it stronger

209 - It might not be individualistic - eg there could be some 'unwritten rules' that teams follow

215-220 - A good point to expand - why is this bad? What is the evidence that inspectors inconsistent in such cases?

251 - Eg this title as a question could be stronger as a statement

253 - assume mean 'how compliance with the requirements'

275+ - I'd make this a point in itself in its own subsection

295 - or this might prompt (ie consumers, as citizens, might call for) further legislation

331+ - Suggest this also becomes a separate point

341 - input requirements generally score higher - they don't a priori, as some could, for example, be based on vague or ambiguous terms. It's just that the ones we use tend to be more reliable (perhaps cos we have weeded out others)

344+ - I am not convinced that outcome-oriented are more related to prevention (though can accept the point about flexibility qua flexibility of inputs). Indeed, could argue that inputs are about prevention - outcomes are only noticed when prevention fails. So either ignore this issue, or make a separate point (maybe in its own section) about private standards (should be) focusing on prevention as much as poss. 

351+ - I had to re-read this paragraph a few times to follow its logic in relation to the paragraph's main point (it seemed a little meandering/stream of consciousness) but maybe I wasnt awake/intelligent enough

376 - Maybe highlight why this is a problem. They could just trust the standards (as people do for Marks and Spencer and RSPCA). And isnt this a problem for legislation too (in democratic legislatures)?

398-9 - This is a reason for valid (or similar term) private standards, not against private standards per se

410+ - re standards vs re audits

414 - standard operators could give anonymised/aggregated data

419 - 'if that'  = missing words?

435+ - great point, well made

447 - 'solely responsible'

492 - meeting volumes doesnt seem directly relevant re standards (except obviously volume of standard-compliant) unless Ive missed something

Author Response

Reviewer: Be clear what is citing earlier papers by the authors
Our response and actions: Actions taken. We have looked through the paper and clarified when and where we cite our previous papers.

Reviewer: I am not sure this paper needs a Materials/Methods section and I don’t think it added much (and, in any case, there is no results section). The paper needs to be located in the literature, in particular the work cited in this section (especially the authors') but I don’t think needs own section
Our response and actions: No actions taken. Since only one of the referee suggested this major change, and not the other, we have decided to keep the Mat/Met section. The paper is mainly based on natural sciences and we believe that having a Mat/Met section in this paper will facilitate for the readers.  

Reviewer: I think the authors' intellects and previous work places them well to give some stronger arguments on some of the issues (where they do, it is really interesting and useful). Often you raise questions/raise issues to consider, but I think you should go further and answer those questions/suggest how to resolve the issues. Indeed, you seem to imply you have a view (sometimes not that subtly), so be braver. I'd personally suggest making it more a series of mini-arguments (and I've suggested some that could be their own sections), each ending on a prescriptive conclusion. It could then be a great paper on how private standards are best used (and/or how not) ie how to fulfil the opportunities and minimize/mitigate the risks (the paper gets to some of these in the conclusion, but do not give the value of these conclusions justice)
The paper feels like (though perhaps not - I haven’t codified it!) it covers risks more than opportunities and could do with a bit of balancing - I think the 'how to do it well' idea suggested above would achieve this.
Our response and actions: We have now tried to be a little bit braver and bolder. We have also clarified and started every section with the opportunities, followed by a couple of risks that need to be thought of if the standard should reach the level of possible opportunities. We have chosen not to have a separate section about opportunities/minimising risks, but instead slightly modified the conclusion to make this such a section. We hope that this will make the reader experience the paper more balanced (between opportunities and risks). However, we also believe that it is important to realise that one big opportunity cannot be overshadowed by a small risk, i.e. it does not necessary need to be the same actual number of opportunities and risks to be balanced. We also find that in the literature (and among politicians and many researchers) the opportunities are already quite well described. The risks to overcome have not been described to the same extent.

Reviewer: - Since the paper is generally comparing private to public (esp in that section), need to be clear where the risks also apply to public legislation
Our response and actions: Actions taken. We agree with the referee and have clarified this.

Reviewer: 44 - maybe specify 'unnecessarily' or 'unavoidably' suffering
Our response and actions: Actions taken in accordance with referee’s suggestion.

Reviewer: 68 - should 'have' be 'has' (or membership become plural)?
Our response and actions: Actions taken in accordance with referee’s suggestion.

Reviewer: 90 - I don’t think regulation is a helpful term here, as might make most readers think only of legal/regulatory settings - whereas the authors mean something wider. Perhaps better just use 'requirement' (as you do) for both
Our response and actions: We have used the current definition of “regulation” in our previous papers about the same topic to cover both legislation and private standards, and also other authors have used it similarly. Therefore, we are reluctant to change the vocabulary now, as this may lead to confusion.  Furthermore, a definition is included in this paper to avoid any possible misunderstanding.

Reviewer: 121+ - I got a bit lost reading this so suggest a bit of a rewrite (depending on above point re line Methods)
Our response and actions: Actions taken in accordance with referee’s suggestion.

Reviewer: 144 - had some requirements (I think)
Our response and actions: Actions taken in accordance with referee’s suggestion.

Reviewer: 149 - Do you have a ref that in these countries legislation/control is suboptimal
Our response and actions: Actions taken. We have rephrased the sentence so it becomes clearer that ref 21 and 22 belongs to the all claims. 

Reviewer: 180-5 - Some greater depth to this point and/or examples would make it stronger
Our response and actions: No actions taken. We really agree that this is a topic of interest, and could be an own paper. However, we have made the decision that more focus on this topic is not within the frames of this paper.

Reviewer: 209 - It might not be individualistic - eg there could be some 'unwritten rules' that teams follow
Our response and actions: Actions taken in accordance with referee’s suggestion.

Reviewer: 215-220 - A good point to expand - why is this bad? What is the evidence that inspectors inconsistent in such cases?
Our response and actions: Actions taken. We have expand the reasoning behind this point.

Reviewer: 251 - Eg this title as a question could be stronger as a statement
Our response and actions: Actions taken in accordance with referee’s suggestion.

Reviewer: 253 - assume mean 'how compliance with the requirements'
Our response and actions: Actions taken. This sentence have been rewritten.

Reviewer: 275+ - I'd make this a point in itself in its own subsection
Our response and actions: No actions taken. We believe that the reasoning is coherent throughout this section. Therefore, we have left it unchanged.

Reviewer: 295 - or this might prompt (ie consumers, as citizens, might call for) further legislation
Our response and actions: Actions taken in accordance with referee’s suggestion.

Reviewer: 331+ - Suggest this also becomes a separate point
Our response and actions: Actions taken in accordance with referee’s suggestion.

Reviewer: 341 - input requirements generally score higher - they don't a priori, as some could, for example, be based on vague or ambiguous terms. It's just that the ones we use tend to be more reliable (perhaps cos we have weeded out others)
Our response and actions: Actions taken in accordance with referee’s suggestion.

Reviewer: 344+ - I am not convinced that outcome-oriented are more related to prevention (though can accept the point about flexibility qua flexibility of inputs). Indeed, could argue that inputs are about prevention - outcomes are only noticed when prevention fails. So either ignore this issue, or make a separate point (maybe in its own section) about private standards (should be) focusing on prevention as much as poss.
Our response and actions: This is what we are trying to say… We have rewritten this somewhat, to make it clearer.

Reviewer: 351+ - I had to re-read this paragraph a few times to follow its logic in relation to the paragraph's main point (it seemed a little meandering/stream of consciousness) but maybe I wasn’t awake/intelligent enough.
Our response and actions: Actions taken. We have tried to make the paragraph clearer.

Reviewer: 376 - Maybe highlight why this is a problem. They could just trust the standards (as people do for Marks and Spencer and RSPCA). And isn’t this a problem for legislation too (in democratic legislatures)?
Our response and actions: We have added the question of trust, and that this issue is a problem in relation to the legislation too.

Reviewer: 398-9 - This is a reason for valid (or similar term) private standards, not against private standards per se
Our response and actions: Actions taken in accordance with referee’s suggestion.

Reviewer: 410+ - re standards vs re audits
Our response and actions: Actions taken in accordance with referee’s suggestion.

Reviewer: 414 - standard operators could give anonymised/aggregated data
Our response and actions: Actions taken in accordance with referee’s suggestion.

Reviewer: 419 - 'if that'  = missing words?
Our response and actions: This sentence has been modified.

Reviewer: 435+ - great point, well made
Our response: Thanks!!

Reviewer: 447 - 'solely responsible'
Our response and actions: Actions taken in accordance with referee’s suggestion.

Reviewer: 492 - meeting volumes doesnt seem directly relevant re standards (except obviously volume of standard-compliant) unless Ive missed something

Our response and actions: Actions taken. We have tried to clarify when this might be a problem.

Reviewer 2 Report

The manuscript aims to analyse opportunities and risks of relying on animal welfare private standards, which is a topic of increasing interest for the majority of farming sectors in many countries. However, the objective is not fully respected and it is missed in some section of the manuscript.

To get the scope, results from two papers by the same authors (Lundmark and co-workers [2] and Lundmark and co-workers [16]) and one unpublished manuscript (Lundmark, F., Hultgren, J. Röcklinsberg, H., Wahlberg, B. &. Berg, C. Non-compliance and follow-up in Swedish official and private animal welfare control of dairy cows, part of the Doctoral Thesis “Lundmark, F. 2016. Mind the Gaps! From Intentions to Practice in Animal Welfare Legislation and Private Standards) are analysed.

Other already published studies (More and co-workers [3] and Main and co-workers [14]) gave a clear and exhaustive overview on the framework of private animal welfare standards and suggested how to examine those schemes. “Material and methods” section states that outcomes from those two papers would have been taken into account. However that part is partially missed in the text and the flow of manuscript is rather hard to follow in some sections.

The discussion deals with the comparison – in terms of intentions, requirements, measurements, results, etc. – between private scheme and EU/national legislation, which is not the main objective as it is described in the aims.

My main suggestion is to revise the aim and the text according to the statement at the end of introduction (lines 81-83). Otherwise it would be interesting for the reader to clarify the comparison and the relationship among some private standards (including outcomes from inspections if available), according e.g. to More and co-workers avoiding the comparison to the national legislation.

Author Response

Reviewer: The manuscript aims to analyse opportunities and risks of relying on animal welfare private standards, which is a topic of increasing interest for the majority of farming sectors in many countries. However, the objective is not fully respected and it is missed in some section of the manuscript.
Our response and actions: The comment from the referee is unclear, however, we have made some changes in the paper, e.g. started each section with the opportunities and developed some reasoning according to referee 1.

Reviewer: To get the scope, results from two papers by the same authors (Lundmark and co-workers [2] and Lundmark and co-workers [16]) and one unpublished manuscript (Lundmark, F., Hultgren, J. Röcklinsberg, H., Wahlberg, B. &. Berg, C. Non-compliance and follow-up in Swedish official and private animal welfare control of dairy cows, part of the Doctoral Thesis “Lundmark, F. 2016. Mind the Gaps! From Intentions to Practice in Animal Welfare Legislation and Private Standards) are analysed. Other already published studies (More and co-workers [3] and Main and co-workers [14]) gave a clear and exhaustive overview on the framework of private animal welfare standards and suggested how to examine those schemes. “Material and methods” section states that outcomes from those two papers would have been taken into account. However that part is partially missed in the text and the flow of manuscript is rather hard to follow in some sections.
Our response and actions: Actions taken. We have referred to More et al. and Main et.al. to a larger extent in the revised version of our paper. We have also tried to clarify some sections and added another subsection in order to make it easier to follow the manuscript. (See also comments to referee 1 about clarifying sentences and sections).  

Reviewer: The discussion deals with the comparison – in terms of intentions, requirements, measurements, results, etc. – between private scheme and EU/national legislation, which is not the main objective as it is described in the aims. My main suggestion is to revise the aim and the text according to the statement at the end of introduction (lines 81-83). Otherwise it would be interesting for the reader to clarify the comparison and the relationship among some private standards (including outcomes from inspections if available), according e.g. to More and co-workers avoiding the comparison to the national legislation.
Our response and actions: Actions taken. We have slightly rewritten the aim in accordance with the referee’s suggestions. The outcomes of different inspections is already discussed in a couple of sections but we have added some more results related to this.

Round 2

Reviewer 2 Report

The manuscript has been improved clarifying aims and referring to previous works (More and co-workers and Main and co-workers) throughtout the text. The manuscript is now suitable for publication in Animals as it is.